# Smart Mn^7+^ Sensing via Quenching on Dual Fluorescence of Eu^3+^ Complex-Modified TiO_2_ Nanoparticles

**DOI:** 10.3390/nano11123283

**Published:** 2021-12-03

**Authors:** Wenbin Yang, Siqi Niu, Yao Wang, Linjun Huang, Shichao Wang, Ketul C. Popat, Matt J. Kipper, Laurence A. Belfiore, Jianguo Tang

**Affiliations:** 1National Center of International Joint Research for Hybrid Materials Technology, National Base of International Sci. & Tech. Cooperation on Hybrid Materials, Qingdao University, 308 Ningxia Road, Qingdao 266071, China; ywb1473259971@163.com (W.Y.); 18348227823@163.com (S.N.); wangyaoqdu@126.com (Y.W.); huanglinjun@qdu.edu.cn (L.H.); wangsc@qdu.edu.cn (S.W.); 2Department of Mechanical Engineering, Colorado State University, Fort Collins, CO 80523, USA; 3School of Biomedical Engineering/School of Advanced Materials Discovery, Colorado State University, Fort Collins, CO 80523, USA; matthew.kipper@colostate.edu; 4Department of Chemical and Biological Engineering, Colorado State University, Fort Collins, CO 80523, USA; belfiore@engr.colostate.edu

**Keywords:** TiO_2_ nanoparticle, Eu(TTA)_3_Phen, fluorescence sensor, Mn^7+^, Eu^3+^

## Abstract

In this work, titania (TiO_2_) nanoparticles modified by Eu(TTA)_3_Phen complexes (ETP) were prepared by a simple solvothermal method developing a fluorescence Mn^7+^ pollutant sensing system. The characterization results indicate that the ETP cause structural deformation and redshifts of the UV-visible light absorptions of host TiO_2_ nanoparticles. The ETP also reduce the crystallinity and crystallite size of TiO_2_ nanoparticles. Compared with TiO_2_ nanoparticles modified with Eu^3+^ (TiO_2_-Eu^3+^), TiO_2_ nanoparticles modified with ETP (TiO_2_-ETP) exhibit significantly stronger photoluminescence under the excitation of 394 nm. Under UV excitation, TiO_2_-ETP nanoparticles showed blue and red emission corresponding to TiO_2_ and Eu^3+^. In addition, as the concentration of ETP in TiO_2_ nanoparticles increases, the PL intensity at 612 nm also increases. When ETP-modified TiO_2_ nanoparticles are added to an aqueous solution containing Mn^7+^, the fluorescence intensity of both TiO_2_ and ETP decreases. The evolution of the fluorescence intensity ratio (I_1_/I_2_) of TiO_2_ and ETP is linearly related to the concentration of Mn^7+^. The sensitivity of fluorescence intensity to Mn^7+^ concentration enables the design of dual fluorescence ratio solid particle sensors. The method proposed here is simple, accurate, efficient, and not affected by the environmental conditions.

## 1. Introduction

Manganese has two primary valence and oxidation states, namely, Mn^2+^ and Mn^7+^, which have different effects in practice [1,2]. Mn^7+^ has been widely used as a strong disinfectant, but its strong oxidation property and its heavy metal characteristic make it a toxic and carcinogenic species in water recycling systems and in human health [3]. In industry, Mn^7+^ has been used as a strong oxidation agent, generating large amounts of toxic waste water [4]. Manganese ions have contributed to serious pollution, causing toxic drinking water and damage to plants [5]. Thus, Mn^7+^ has attracted a lot of attention among pollutants in recent years, and the detection of Mn^7+^ is very important for environmental protection. Ion chromatography (IC) [6], atomic absorption spectroscopy (AAS) [7], inductively coupled plasma mass spectroscopy (ICP-MS) [8] and spectrophotometry [9] can be used to detect Mn^7+^. Monitoring Mn^7+^ in water samples requires complex methods such as atomic spectrometry. However, due to the low efficiency of this method and the interference of impurities present in the real samples, the detection of Mn^7+^ at low concentration is complicated and requires pretreatment steps [10]. Therefore, it is necessary to explore a simple, accurate, efficient, and low-interference method to detect Mn^7+^ in complex samples.

Even with the use of highly sensitive metal ion detectors, the content level of Mn^7+^ in environmental samples is usually low or close to the detection limit, so the samples require a unique separation. When high concentrations of interfering particles are present in the matrix, efficient extraction of Mn^7+^ is required. Qian et al. proposed a FAAS method that uses crosslinked chitosan to separate Mn^2+^ and Mn^7+^. This method is simple and sensitive and can be used for environmental sample detection [11]. A potential problem limiting the application of this method for actual samples is that the Mn^2+^ is determined by its oxidation to Mn^7+^, and then the total Mn concentration is determined. Therefore, this method requires that the analyte species are only Mn^2+^ and Mn^7+^. This motivates the development of new Mn sensors.

Europium complexes constitute an important class of optical probes, with applications ranging from sensing of bioactive species, high throughput assays and screening protocols in vitro, to time-resolved imaging studies in cellulo or in vivo [12]. Eu complexes might also provide an opportunity for Mn^7+^ sensing. In this work, we design a strategy to combine the advantages of Eu^3+^ fluorescence and TiO_2_ dispersion in water. Our fluorescent nanomaterials can detect in water or other liquids, with minimal interference and low cost. In recent years, fluorescence spectrometry has been used to detect and quantify metal ions [13]. A new method for the determination of copper, manganese, nickel, and lead in diesel oil has been developed, which combines liquid–liquid reversed-phase eddy microextraction and energy dispersive X-ray fluorescence spectroscopy [14]. Fluorescence sensors [15] have received widespread attention due to their high sensitivity, selectivity, and simple operation; however, a single wavelength fluorescence sensor is still affected by sample concentration, environmental factors, and excitation intensity [16]. Dual fluorescence sensors can measure emission peaks at two different wavelengths, and use the ratio of the two peak intensities to solve the above problems [17,18,19,20], increasing sensitivity and selectivity [21,22]. Nanomaterial fluorescence sensors are a new type of sensors, which have large specific surface area, controllable size, predictable nanostructure [23,24,25], and polychromatic and adjustable radiation characteristics [26]. In general, photoluminescence is achieved by organic lanthanide complexes that absorb UV light and emit photons due to f-f or f-d electronic transitions in the lanthanide ion [27,28]. Among all the lanthanide ions, europium ions have been widely used because of their prominent emission peak and long fluorescence lifetime [29,30,31]. The advantages of their complexes are stable characteristic emission peaks and increased fluorescence intensity. TiO_2_ nanoparticles have excellent optical properties, catalytic properties, chemical stability, and biocompatibility [32]. Therefore, TiO_2_-ETP nanoparticles combine the advantages and fluorescence properties of europium complexes and TiO_2_ nanoparticles.

In this work, we synthesized europium complex-modified TiO_2_ nanoparticles (TiO_2_-ETP) by a solvothermal method, and we investigated the structure, properties, and application of TiO_2_-ETP nanoparticles. The resulting nanoparticles have outstanding luminescence characteristics, indicating the possibility of using TiO_2_ nanoparticles to improve the effective luminescent properties of rare earth complexes. TiO_2_-ETP nanoparticles exhibit significantly stronger photoluminescence (PL) than TiO_2_-Eu^3+^ nanoparticles. Thus, TiO_2_-ETP nanoparticles have the potential to be used as new semiconductor luminescent materials. In our study, the fluorescence intensity of TiO_2_-ETP was sensitive to the change of Mn^7+^ concentration. In addition, the high specific surface area of TiO_2_ nanoparticles can increase the contact area between the sensor and Mn^7+^, which can improve the sensitivity of the sensor. In the fluorescence spectrum, excited with 394 nm light, TiO_2_-ETP has emission peaks at 454 nm and 616 nm for titania and the ETP, respectively. The fluorescence of ETP and TiO_2_ both decreases in the presence of Mn^7+^, but with different characteristic sensitivity to Mn^7+^. The Mn^7+^ dual fluorescence sensor shows a wide detection range and high sensitivity, and the effectiveness of the sensor has been verified through experiments. In this research, we propose an intelligent dual fluorescence sensor, which is low-cost and easy to operate. It has high sensitivity and high efficiency. Compared with previous reports [33,34], our method is simple and practical, reduces the need for pretreatment, and has a larger detection range. The preparation and detection mechanism of the sensor is shown in Figure 1.

## 2. Experimental Details

Ethanol (AR, 99.7%), acetic acid (AR, 99.7%), and tetrabutyl titanate (AR) were purchased from Macklin (Shanghai, China). Europium chloride hexahydrate (EuCl_3_·6H_2_O, 99.9%), 1,10-phenanthroline monohydrate (Phen, AR, 98%) and 2-thenoyltrifluoroacetone (TTA, 98%) were purchased from Aladdin (Shanghai, China).

As shown in Figure 2, TiO_2_, TiO_2_ modified with Eu^3+^ (TiO_2_-Eu^3+^), and TiO_2_ modified with Eu (TTA)_3_Phen (ETP) (TiO_2_-ETP) were prepared using the solvothermal method. Tetrabutyl titanate (TBT) was used as a precursor. Ethanol (CH_3_CH_2_OH) and acetic acid (CH_3_COOH) were used as solvents and hydrolysis inhibitors. Before the final synthesis, two solutions were prepared (solution A and solution B). Solution A was prepared by adding acetic acid and TBT in ethanol. In solution B, EuCl_3_ was dissolved in ethanol by stirring. Then 1,10-phenanthroline monohydrate and methyl 1H-benzotriazole, dissolved in absolute ethanol were added to solution B, and the mixed solution was stirred with a magnetic stirrer for 1 h at room temperature. Solution A was added to solution B. The mixture became cloudy with continuous stirring. The mixture was heated in an autoclave to 150 °C for 24 h. After the reaction, the resulting material was cooled to room temperature. The synthesized material was centrifuged and washed with ethanol and distilled water several times to remove impurities. The resulting white solid was collected and dried in an oven at 60 °C. For the synthesis of unmodified TiO_2_ nanoparticles, the same conditions are used, without the addition of ETP.

The quenching experiments using metal ions were performed by adding TiO_2_-ETP (0.1 mol/L) into different metal ion analyte solutions with the concentrations of 1 mM/L, and the mixtures were stirred for 2 h. To determine the quenching behavior, Mn^7+^ concentrations in the range of 1 µM/L to 1000 µM/L were used.

A Thermo Scientific F200i (Thermo, Waltham, MA, USA) transmission electron microscope was used to obtain transmission electron microscopy (TEM) images at an accelerating voltage of 200 kV. X-ray powder diffraction (XRD) measurements were performed using a Bruker D8 Advance diffractometer (Bruker, Karlsruhe, Germany), which was operated at a generator voltage of 40 keV and a current of 30 mA. The X-ray source is CuKα radiation (λ = 0.154 nm). The diffraction pattern was collected at a scanning speed of 1°/min within a 2θ scanning range of 20° to 80°. Measurements of Raman spectra were performed on a Thermo Scientific DXR 2xi (Thermo, Waltham, MA, USA) Raman Spectrometer under a backscattering geometry. The valence states of Eu, O, and Ti atoms were measured by X-ray photoelectron spectroscopy (XPS) on a Thermo Scientific ESCALAB 250 (Thermo, Waltham, MA, USA) spectrometer. The XPS experiment was performed under vacuum using AlKα (1486.6 eV) radiation. The ultraviolet absorption spectrum was obtained using PerkinElmer Lambda 750s (PerkinElmer, Shanghai, China) with a solid sample frame, on which the powder samples were flattened when the powder samples were used. The PL spectrum is an important tool for determining the luminescent properties of materials. An Edinburgh Instrument Fluorescence Spectrometer FLS 1000 (Livingston, Edinburgh, UK) was used to record the excitation and emission spectra of each sample, on which the data of excitation spectra, emission spectra, fluorescence lifetimes were collected. A 450W xenon arc lamp capable of emitting a continuous spectrum with greater intensity was used as the light source. The excitation monochromator was used to select the specified spectrum with the excitation wavelength of 394 nm. Fluorescence analyzer calibration was performed in accordance with the instrument operating procedures using standard sample, sample preparation and processing, resulting in excellent calibration curves.

## 3. Results and Discussion

### 3.1. Morphological Structures

The additions of Eu^3+^ and ETP into TiO_2_ change the shape and size of TiO_2_ nanoparticles. Figure 3 shows typical transmission electron microscopy (TEM) images of TiO_2_ nanoparticles. TiO_2_ nanoparticles with spherical morphology can be seen in TEM images (Figure 3a). The morphology of TiO_2_ nanoparticles with Eu^3+^ varies from spherical to ellipsoidal shapes (Figure 3b). The TiO_2_ nanoparticles modified with ETP have a cuboid shape (Figure 3c). These changes are similar to the previous report [35]. Eu^3+^ and ETP-doped TiO_2_ cause different shapes of TiO_2_-Eu^3+^ and TiO_2_-ETP nanoparticles [36]. The corresponding histograms of the diameter distributions and the changes of the average sizes are shown in Figure 3 in which the average nanoparticle sizes can be found to be 15 ± 0.09 nm, 12.3 ± 0.08 nm, and 9 ± 0.1 nm in diameter. The length of TiO_2_-Eu^3+^ is between 10 and 40 nm. Compared to TiO_2_, the average size of TiO_2_-ETP nanoparticles decreases, which suggests that the inclusion of ETP largely suppresses the growth of TiO_2_ nanoparticles. This size change of TiO_2_-Eu^3+^ and TiO_2_-ETP nanoparticles can also relate to crystalline structures described later, based on X-ray diffraction analyses [37]. The growth of TiO_2_-Eu^3+^ crystals is hindered by the formation of Eu-O-Ti bond in the crystal void of TiO_2_-Eu^3+^ nanoparticles. The decrease of particle size of ETP-modified TiO_2_ is mainly caused by ETP entering the lattice and binding with oxygen. Due to internal stress in the crystal lattice, the diffusion of Ti^4+^ and O^2-^ and the obstacle of crystal migration, the crystal growth at the boundary is retarded [38].

### 3.2. Crystalline Structure

Modification with Eu^3+^ can effectively change the crystal structure and inhibit grain growth of TiO_2_ nanoparticles. This effect is more pronounced when the organic complex (ETP) is used. Figure 4 shows the diffraction patterns of TiO_2_ nanoparticles obtained by the solvothermal method. The presence of diffraction peaks corresponding to (101), (004), (200), (105), (211), and (204) planes indicate the formation of the anatase TiO_2_ phase [39]. The XRD shows that TiO_2_-Eu^3+^ and TiO_2_-ETP nanoparticles have peaks at 2θ = 25.3°, 38.1°, 47.9°, 54.1°, 55.2°, and 62.6°, which correspond to peaks of anatase TiO_2_ (JCPDS NO.21-1272). No additional peaks of any other phases or impurities were found, which indicates the high purity of the nanoparticles. Figure 4 shows that the XRD peaks of the (101) crystal plane in TiO_2_-ETP are slightly shifted towards a smaller diffraction angle from 25.3° to 25.1°, while other diffraction peaks have almost no observable shift. This is likely due to the addition of ETP [40]. Because the smaller diffraction angle relates to the larger gaps between crystal planes, this shift means that the distance of the (101) crystal plane slightly increases upon ETP addition [40]. The relative intensity of the peak at 2θ = 25.3° is significantly decreased in TiO_2_-ETP compared to the TiO_2_ and TiO_2_-Eu^3+^ nanoparticles, indicating that the crystallinity decreased [41]. When ETP is added to TiO_2_ nanoparticles, deformation is induced in the system, leading to a change in the periodicity of the lattice and a decrease in the crystal symmetry. From the full width at half maximum, one can judge that TiO_2_-ETP has a smaller particle size than TiO_2_ and TiO_2_-Eu^3+^. The characteristic peaks of the (101) (004), and (200) crystal planes from the XRD image were selected, and the Scherrer formula (Equation (1)) was used to calculate the average size of the modified and unmodified nanoparticles (Table 1),
(1)$$L_{hkl}=\frac{K\lambda }{\beta \cos\theta },$$
where $L_{hkl}$ is the size of the particle crystallites, K is the shape constant, usually taken as 0.9, λ is the wavelength of X-rays (CuKα is 1.5406 Å), β is the full diffraction width at half maximum, measured in radians at 2θ Peak.

Figure 5 shows the Raman spectra of the obtained TiO_2_ nanoparticles. The Raman peaks at 143, 395, 514, and 639 cm^−1^ correspond to E_g_, B_1g_, A_1g_, or B_1g_, and E_g_ of the anatase phase, respectively [42]. The most dominant E_g_ mode appears due to the external vibration of the anatase structure at 143 cm^−1^. This indicates that the anatase phase is formed in the prepared europium complex-modified TiO_2_ nanoparticles. The inclusion of ETP in TiO_2_-ETP nanoparticles changes features of the crystal structure of TiO_2_, so the Raman spectrum was slightly shifted. It can be seen from the Raman spectrum that, especially in the E_g_ mode near 144 cm^−1^, the TiO_2_ nanoparticles modified with ETP move to a higher wavenumber direction, and their intensity drops sharply. The observation can be explained by a decrease in the particle size in TiO_2_-Eu^3+^ [41,43,44]. When the grain size decreases, it will significantly affect the Raman spectrum of titanium dioxide nanoparticles. Generally speaking, dimensional changes will produce pressure, and volume shrinkage will occur in TiO_2_ nanoparticles. The reason for the increase in pressure is the decrease in the distance between atoms. The sudden drop in the intensity of the Raman spectrum, especially the drop in the scattering intensity of the E_g_ mode, is related to the destruction of the atomic symmetry of the crystal, which is caused by the defects modified with ETP. Because TiO_2_-ETP nanoparticles have local lattice defects, the Raman peak becomes weaker and broader, which means that the crystallinity of synthesized nanoparticles decreases.

### 3.3. Confirmation of Eu^3+^ in TiO_2_

X-ray photoelectron spectroscopy (XPS) was used for elemental analysis of ETP-modified titanium dioxide nanoparticles. Figure 6A(a–c) shows the survey XPS spectra of TiO_2_, TiO_2_-Eu^3+^, and TiO_2_-ETP, respectively. The XPS spectra in Figure 6B clearly shows the changes of the binding energy of the Ti2p electron orbital in TiO_2_, TiO_2_-Eu^3+^, and TiO_2_-ETP in which the binding energies in TiO_2_, TiO_2_-Eu^3+^ and TiO_2_-ETP, are gradually decreased at 458.72, 458.62, and 458.57 eV. This phenomenon is similar to a previous report [45]. The binding energy decreases are caused by Eu^3+^ and ETP inserting between crystal planes. The much larger decrease of binding energy in TiO_2_-ETP is due to the larger TTA and Phen ligands carried by Eu^3+^. Figure 6C shows spectra of Eu3d with significantly higher intensity for TiO_2_-ETP than for TiO_2_-Eu^3+^, indicating that the TTA and Phen ligands tightly bind the Eu. At the same time, the binding energy of Eu3d in TiO_2_-ETP is slightly lower than that in TiO_2_-Eu^3+^, which is also due to the stronger interaction of ligands with the Eu3d electron orbital [46]. Figure 6D–F show the binding energy changes of O1s in TiO_2_, TiO_2_-Eu^3+^, and TiO_2_-ETP, showing that Ti-O and Eu-O have almost the same binding energies in TiO_2_-Eu^3+^ and TiO_2_-ETP. The binding energy of Ti-O in TiO_2_ is higher (Figure 6D) than the corresponding binding energy in the TiO_2_-Eu^3+^ (Figure 6E) and TiO_2_-ETP (Figure 6F), indicating the influence of Eu^3+^ insertion between crystal planes of TiO_2_. The binding energies corresponding to Ti-O, O-H, Eu-O, and C=O in TiO_2_-ETP (Figure 6F) are located at 529.8, 530.8, 531.4, and 532.3 eV. Compared to TiO_2_, the formation of an Eu-O bond indicates that Eu has reacted with TiO_2_. The C=O bond belongs to TTA in ETP, which indicate that ETP is interacting with TiO_2_ [47,48,49].

### 3.4. UV Absorption and Bandgap of TiO_2_

Figure 7a,c show the UV-visible absorption curves of TiO_2_-Eu^3+^ and TiO_2_-ETP. Compared with the curves of TiO_2_-Eu^3+^ (Figure 7a), the curves of TiO_2_-ETP have significant redshift (Figure 7c). As the ETP concentration increases, the absorption edge moves to the right, and the energy required to generate electron-hole pairs gradually decreases. The valence band of TiO_2_ absorbs ultraviolet light and releases it into the conduction band and defect state energy level of TiO_2_. Because the excited state of Eu^3+^ is lower than the conduction band and defect state, the energy is transferred to Eu^3+^ [50]. UV–visible spectra shown in Figure 7b show that modification with ETP shifted the TiO_2_ absorption edge from the UV to the visible region. This means Eu^3+^ and ETP doping produce defects in the TiO_2_ host crystal, and thus these defects result in band gap decrease [51,52,53].

The absorption spectra in the UV and visible regions of TiO_2_-Eu^3+^ and TiO_2_-ETP nanoparticles are used to estimate the bandgap. [F(R) × hν] ^1/2^ of hν in the vicinity of the absorption edge are plotted for all samples in Figure 7c,d, where F(R) is the Kubelka–Munk function, defined as F(R) = (1 − R)^2^/2R, hν is the photon energy, and R is the reflection coefficient converted to absorption intensity. By extrapolating the linear part of the curve to the intersection with the *x*-axis, the bandgap energies can be estimated. The bandgap energies for TiO_2_-Eu^3+^ are 3.08 eV, 3.05 eV, 3.06, and 3.09 eV for TiO_2_ modified by 2, 4, 6, and 8 mol% Eu^3+^, respectively. The bandgap energies for TiO_2_-ETP are 2.72 eV, 2.40 eV, 2.30 eV, and 2.26 eV for TiO_2_ modified by 2, 4, 6, and 8 mol% ETP, respectively. Compared with the band gap of 3.2 eV of TiO_2_, the band gaps of TiO_2_-Eu^3+^ and TiO_2_-ETP are decreased. The bandgap decrease is caused by interactions between TiO_2_ host and dopants, either Eu^3+^ or ETP in TiO_2_-Eu^3+^ and TiO_2_-ETP, respectively. Based on the results of XRD to indicating the (101) crystal plane distance extension, and the XPS to confirm the interactions between Eu^3+^-O and the binding energy changes of Eu3d and Ti2p in Eu^3+^ and ETP, we conclude that Eu^3+^ and ETP as dopants have interacted with TiO_2_ in different ways. Thus, these changes can be attributed to the “solubility limit” of Eu^3+^ and ETP in TiO_2_ host. The former Eu^3+^ is from EuCl_3_·6H_2_O in which both the Cl^−^ counter ion and the bound H_2_O molecules affect the solubility of Eu^3+^. However, ETP is a complex with the organic ligand molecules (Phen and TTA), which modify the solubility of ETP. Solubilized ions have efficient interaction with the TiO_2_ host to change the electron transition bandgap [54]. These interactions also affect the fluorescence behaviors as shown in Figure 8.

### 3.5. Photoluminescence Analysis

The luminescence mechanism of Eu^3+^-complexes is generally described as follows: the organic ligand absorbs incident photons, transitioning from the ground state to the excited singlet state. Normally, the excited electron will experience an intersystem transition from the singlet state to the triplet state. The triplet excited state transfers energy to the S1 excited state of Eu^3+^, which can subsequently emit a photon when the ^5^D_0_ transitions to the ^7^F_J_ configurations. The luminescence of TiO_2_ is due to the electron transition between the valence band and the conduction band. Figure 8a shows the excitation spectra of TiO_2_-Eu^3+^ and TiO_2_-ETP nanoparticles. The excitation spectra are measured by the emission wavelength of the nanoparticles at 612 nm. The characteristic excitation peak is related to the 4f-4f transition of Eu^3+^ from ^7^F_0_. The excitation spectrum consists of sharp lines at 384, 394, 418, and 464 nm, assigned to the ^7^F_0_→^5^L_7_, ^7^F_0_→^5^L_6_, ^7^F_0_→^5^D_3_, and ^7^F_0_→^5^D_2_ transitions of Eu^3+^ [55]. Strong peaks at 394 nm and 464 nm correspond to the ^7^F_0_→^5^L_6_ and ^7^F_0_→^5^D_2_ Eu^3+^ transitions. The intensity of the excitation spectrum of TiO_2_-ETP is higher than that of TiO_2_-Eu^3+^. The organic ligands in ETP help absorb more ultraviolet light. Figure 8b shows the emission spectra of TiO_2_-Eu^3+^ and TiO_2_-ETP. When excited at a wavelength of 394 nm, the emission spectrum consists of ^5^D_0_→^7^F_J_ (J = 0, 1, 2, 3, 4) (578, 592, 612, 652, and 703 nm) Eu^3+^ transitions. Due to the allowable electric dipole of the ^5^D_0_→^7^F_2_ transition, the strongest emission is produced at 612 nm, which is red. Figure 8c shows the emission spectra of TiO_2_-Eu^3+^, prepared with different concentrations of Eu^3+^ (2, 4, 6, and 8 mol%). The influence of concentration on PL intensity is shown in Figure 8e. The optimal concentration of Eu^3+^ is 4% [39]. When the concentration exceeds 4%, the fluorescence of TiO_2_-Eu^3+^ nanoparticles decreases. This suggests that 4% Eu^3+^ concentration is the upper solubility limit in the TiO_2_ host. However, the fluorescence intensity of TiO_2_-ETP increases with increasing concentrations of ETP. This indicates that the organic ligands in ETP improve the solubility of ETP in the TiO_2_ host, which provide a more effective “antenna effect” of organic ligands [56,57,58].

Figure 8f shows a diagram of energy levels. Based on XRD, Raman, and XPS analysis, Eu^3+^ and ETP were successfully incorporated into TiO_2_ nanoparticles. In Figure 8f, the phrase “defect state” is representative of a variety of defects. Europium ions and ETP will produce point defects in the crystal lattice and combine with oxygen atoms to form Eu-O bonds [59], and the multiple defect energy levels are marked as multiple lines. This indicates that the external ultraviolet rays are absorbed by the TiO_2_ nanoparticles, and the energy enters the defect state. Energy is then transferred to the Eu in the ETP, realizing the energy transfer process from TiO_2_ to Eu. Because the energy level of the emission state of Eu^3+^ is lower than the energy level of the defect in TiO_2_ nanoparticles, the energy is transferred from the defect state of TiO_2_ to the crystal field state of Eu^3+^ ions, which leads to effective photoluminescence of the nanoparticles. Due to the small size and a large number of nanoparticles, there are many surface states available for transferring energy to the states of the crystal field of Eu^3+^. Figure 9 shows the fluorescence lifetime diagram of TiO_2_-Eu^3+^ and TiO_2_-ETP. The fluorescence attenuation of TiO_2_-ETP is slower than that of TiO_2_-Eu^3+^, and the quantum yield of TiO_2_-ETP is higher than that of TiO_2_-Eu^3+^. The fluorescence lifetime of TiO_2_-ETP and TiO_2_-Eu^3+^ were 0.51 ms and 0.39 ms, and the quantum yields of TiO_2_-ETP and TiO_2_-Eu^3+^ were 10% and 5%.

### 3.6. Fluorescence Spectra of TiO_2_-ETP in the Presence of Metal Ions

Eu^3+^ can be complexed with organic ligands containing oxygen or nitrogen groups, such as methyl 1H-benzotriazole and 1,10-phenanthroline monohydrate [60,61]. Therefore, when the europium complex is in contact with metal ions, the fluorescence properties will change. In this paper, common metal cations such as Zn^2+^, Mn^3+^, K^+^, Mn^7+^, Fe^2+^, Mg^2+^, Ca^2+^, and Co^2+^ are selected to determine whether these metal ions will affect the fluorescence properties of TiO_2_-ETP. These experimental analyses prove that these common impurities will not affect the sensitivity of the sensor. The results of these experiments are shown in Figure 10. We have also previously reported of the effects of organic molecules, such as carbohydrates, cholesterol, and amino acids, on the emission of Eu^3+^ complex in different hosts, showing that the tested organic molecules exhibit no quenching effect [40,62].

As shown in Figure 10a, TiO_2_-ETP shows a strong fluorescence peak located at 464 and 616 nm with excitation at λ_ex_ = 394 nm. The fluorescence of TiO_2_-ETP is influenced by the addition of Mn^7+^, where a significant quenching effect can be observed. The fluorescence intensity of ETP-modified TiO_2_ nanoparticles decreases with the increase of Mn^7+^ concentration in the solution. Figure 10b shows the ratio (I/I_0_) of the fluorescence intensity of TiO_2_-ETP in an aqueous solution containing no metal ions and a solution containing a single metal ion. I_0_ is the fluorescence intensity of TiO_2_-ETP in the absence of metal ions at 464 nm and 616 nm, and I is the fluorescence intensity of TiO_2_-ETP at 464 nm and 616 nm in the presence of a single metal ion. It can be seen from Figure 10b that the addition of other metal ions besides Mn^7+^ will not significantly reduce the fluorescence intensity of TiO_2_-ETP. The aqueous solution containing Mn^7+^ will cause fluorescence quenching of TiO_2_-ETP. The decrease in fluorescence intensity can also be detected by adding TiO_2_-ETP to an aqueous solution containing a small amount of Mn^7+^. Experiments show that when other ions are present, only manganese will quench the Eu^3+^ fluorescence. The possible mechanism of quenching can be either the absorption of photons by Mn^7+^, or the Mn^7+^ excimer formation by interaction with the excited state of with ETP, preventing energy transfer to Eu^3+^. The detection of Mn^7+^ at the micromolar level can be achieved. Based on the different responses of TiO_2_-ETP in the presence of Mn^7+^ and other metal ions, a method is proposed for determining the concentration of Mn^7+^.

Figure 10c,e shows the relationship between the fluorescence intensity of TiO_2_-ETP and the concentration of Mn^7+^ in an aqueous solution. For semiconductor TiO_2_-ETP, fluorescence quenching is explained by the efficient electron transition process through annihilation of nonradiative electron-hole recombination. The quenching normally is from the Mn^7+^ acceptance of energy from the excited states of TiO_2_-ETP. Because there are two excited states corresponding to TiO_2_ and ETP, the emissions of TiO_2_ and ETP will be quenched by Mn^7+^. The Stern–Volmer diagram used to determine the sensitivity of Mn^7+^ to TiO_2_-ETP is shown in Figure 10d,f. The Mn^7+^ concentration is linearly related to the fluorescence intensity. As the concentration of Mn^7+^ increases from 0 μmol/L to 1 mmol/L, the position of the fluorescence emission peak does not move, and the fluorescence intensity of TiO_2_-ETP gradually decreases. This linear relationship means that the charge transfer mechanism between Mn^7+^ and TiO_2_-ETP is caused by a dynamic mechanism. Figure 10d shows a graph of the variation of the radiation intensity (I/I_0_) of TiO_2_ at 464 nm as a function of the concentration of Mn^7+^. A linear regression equation is obtained: I/I_0_ = 20.7C + 23,570.9 with a correlation coefficient R^2^ equal to 0.99 (*n* = 14), where I_0_ is the TiO_2_-ETP radiation intensity at 464 nm, I is the intensity of TiO_2_-ETP with different concentration of Mn^7+^, and C is the concentration of Mn^7+^. Likewise, Figure 10f shows a graph of the variation of the emission intensity (I/I_0_) of TiO_2_-ETP at 616 nm as a function of the Mn^7+^ concentration. The linear regression equation for Mn^7+^ is I/I_0_ = 13.8C + 2882.3 (R^2^ = 0.98, *n* = 14).

## 4. Conclusions

In this study, we have synthesized TiO_2_-ETP nanoparticles using a simple solvothermal process. XRD patterns, Raman spectra, and XPS spectra show that ETP is successfully incorporated into TiO_2_ nanoparticles. TiO_2_-ETP nanoparticles exhibit a higher PL intensity than TiO_2_-Eu^3+^ nanoparticles upon excitation at a wavelength of 394 nm. With the increase of Eu^3+^ concentration, the fluorescence intensity of TiO_2_-Eu^3+^ at 550–750 nm increases, and the optimal concentration is 4.0 mol%. When the concentration of Eu^3+^ exceeds 4.0 mol%, the fluorescence decreases, indicating that a solubility limit has been reached. TiO_2_-ETP overcomes the solubility limit, and realizes a fluorescence increase with increasing ETP concentration. Exploiting the quenching effect of Mn^7+^ on the fluorescence intensity of TiO_2_-ETP, a simple and efficient Mn^7+^ fluorescence sensor was proposed. Unlike the previously reported detection using Eu (TTA)_3_Phen or TiO_2_, the detection range of the TiO_2_-ETP nanomaterial is larger, and the detection accuracy and sensitivity are higher. Experimental results show that the proposed new sensor is practical, can be used to detect real samples, does not exhibit interference with common metal ions, can be used for detection in complex environments, is simple to operate, and has excellent potential for application.

## Figures and Tables

**Figure 1 nanomaterials-11-03283-f001:**
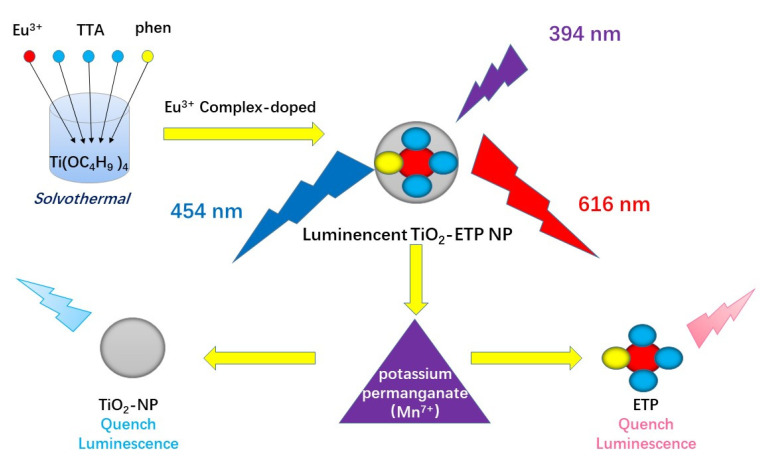
Schematic illustration of TiO_2_ nanoparticles modified with Eu(TTA)_3_Phen preparation and sensing mechanism of manganese ion concentration.

**Figure 2 nanomaterials-11-03283-f002:**
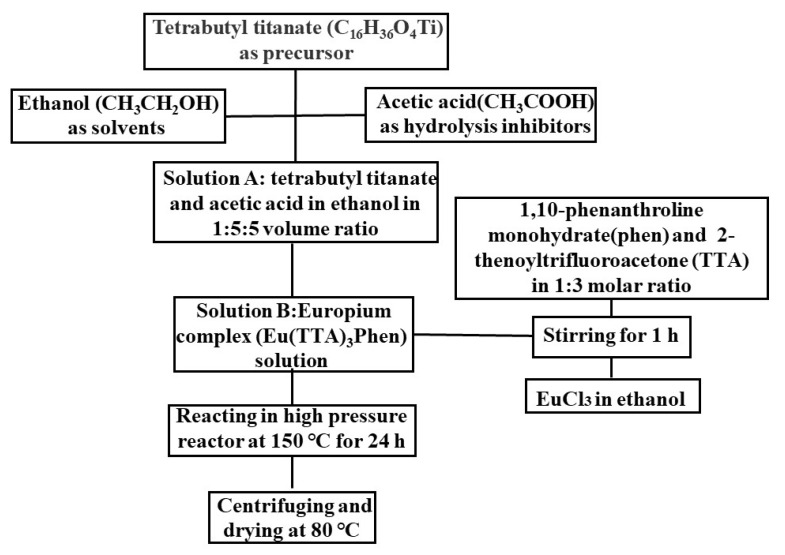
Flow chart of TiO_2_-ETP preparation.

**Figure 3 nanomaterials-11-03283-f003:**
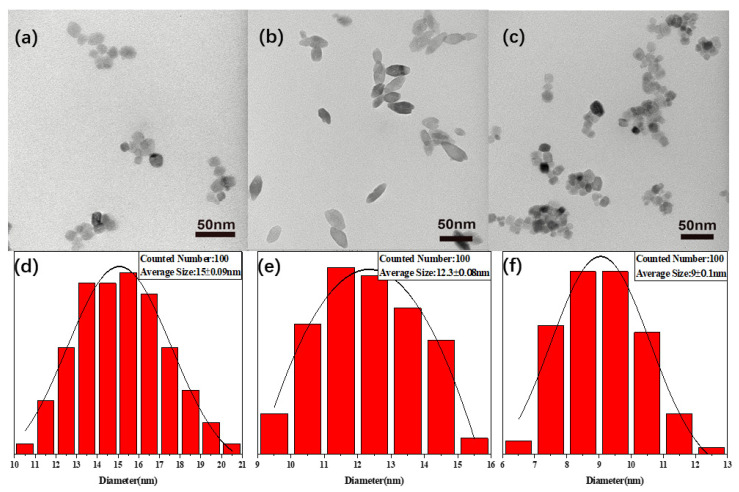
TEM images of (**a**) TiO_2_ (**b**) TiO_2_-Eu^3+^ (**c**) TiO_2_-ETP and diameter distribution histograms of (**d**) TiO_2_, (**e**) TiO_2_-Eu^3+^, (**f**) TiO_2_-ETP.

**Figure 4 nanomaterials-11-03283-f004:**
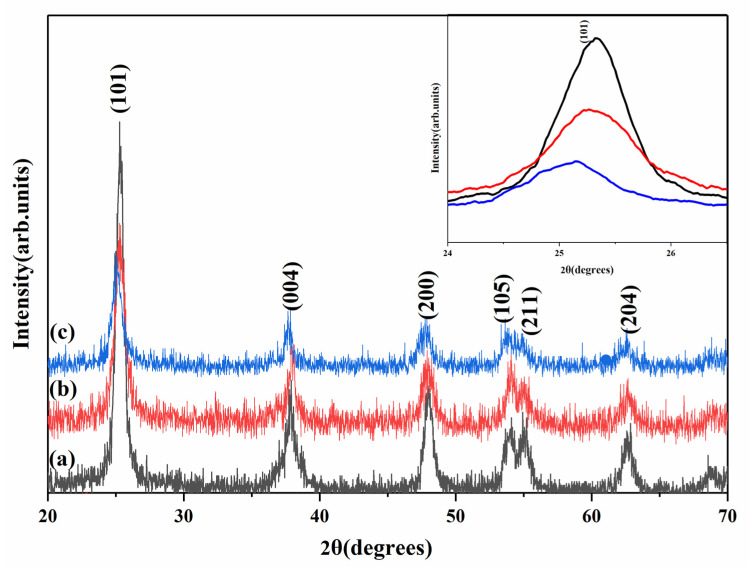
XRD patterns for (**a**) TiO_2_, (**b**) TiO_2_-Eu^3+^, (**c**) TiO_2_-ETP.

**Figure 5 nanomaterials-11-03283-f005:**
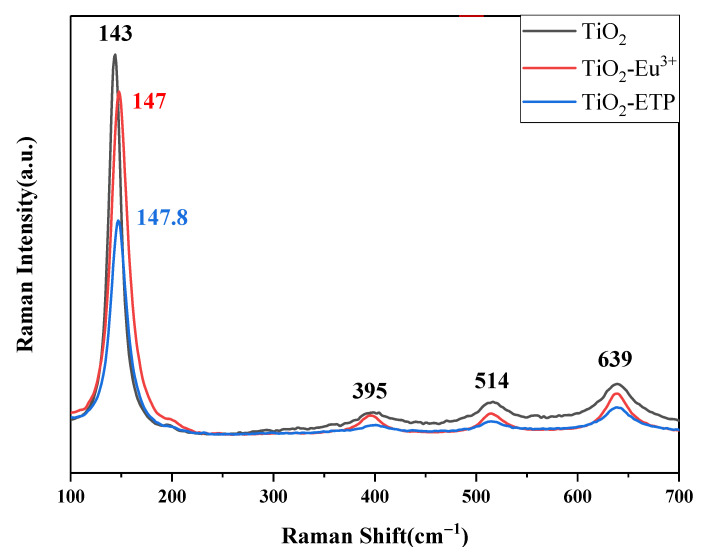
Raman spectra of TiO_2_, TiO_2_-Eu^3+^, and TiO_2_-ETP.

**Figure 6 nanomaterials-11-03283-f006:**
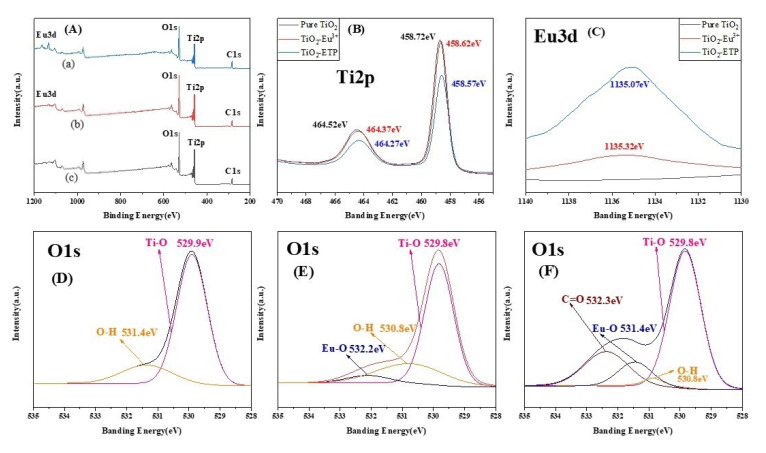
(**A**) Survey XPS spectra, (**B**) XPS spectra of Ti2p, (**C**) XPS spectra of Eu3d, (**D**) XPS spectra of O1s in TiO_2_, (**E**) XPS spectra of O1s in TiO_2_-Eu^3+^, (**F**) XPS spectra of O1s in TiO_2_-ETP.

**Figure 7 nanomaterials-11-03283-f007:**
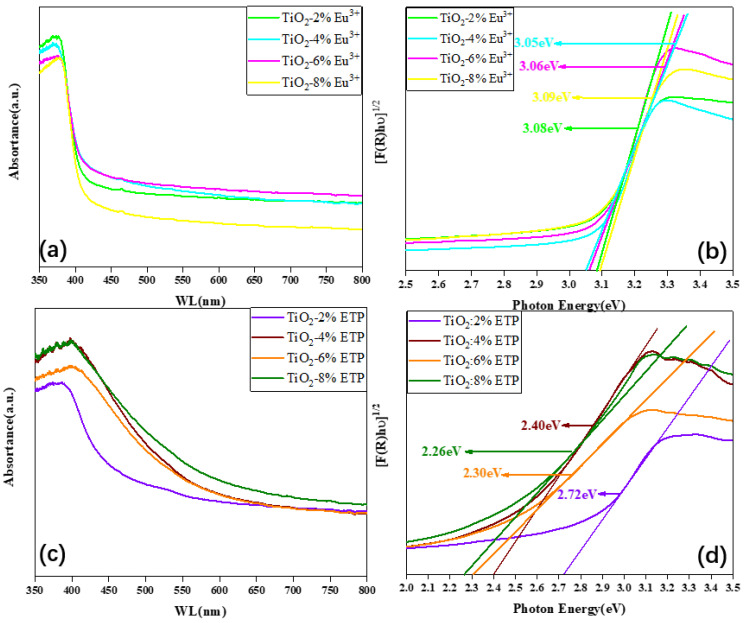
UV-Vis spectra (**a**,**c**) of TiO_2_-Eu^3+^ and TiO_2_-ETP nanoparticles prepared with different amounts of the dopant (2%, 4%, 6%, and 8%). Kubelka-Munk function for band gap estimation (**b**,**d**) of TiO_2_-Eu^3+^ and TiO_2_-ETP.

**Figure 8 nanomaterials-11-03283-f008:**
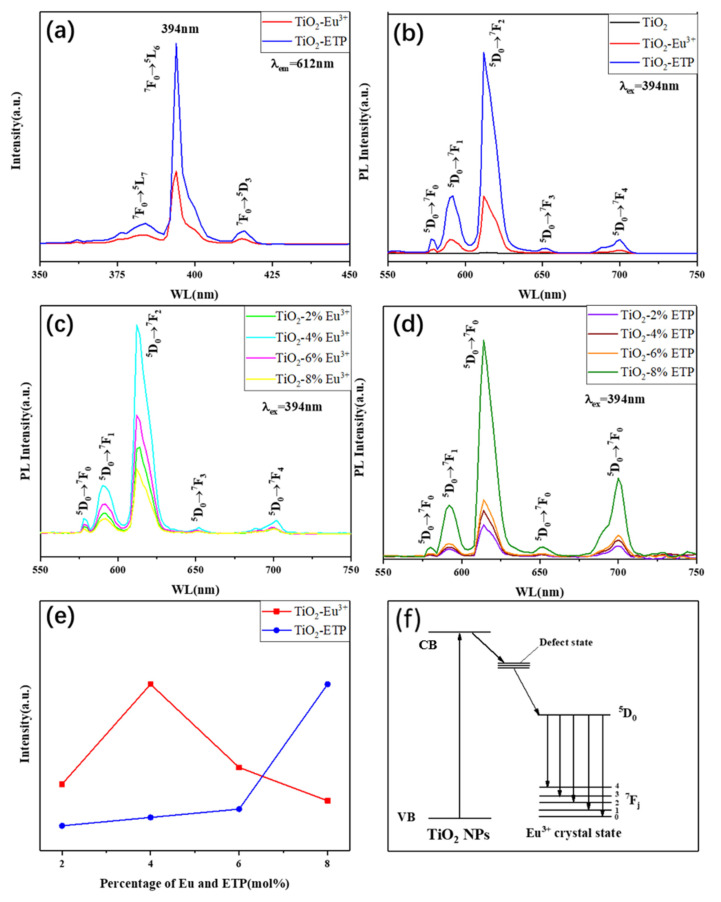
The photoluminescence excitation spectra (**a**) and emission spectra (**b**–**d**) of TiO_2_, TiO_2_-Eu^3+^ and TiO_2_-ETP nanoparticles; fluorescence intensity vs. concentration curve (**e**) and energy transfer diagram (**f**) of TiO_2_, TiO_2_-Eu^3+^ and TiO_2_-ETP.

**Figure 9 nanomaterials-11-03283-f009:**
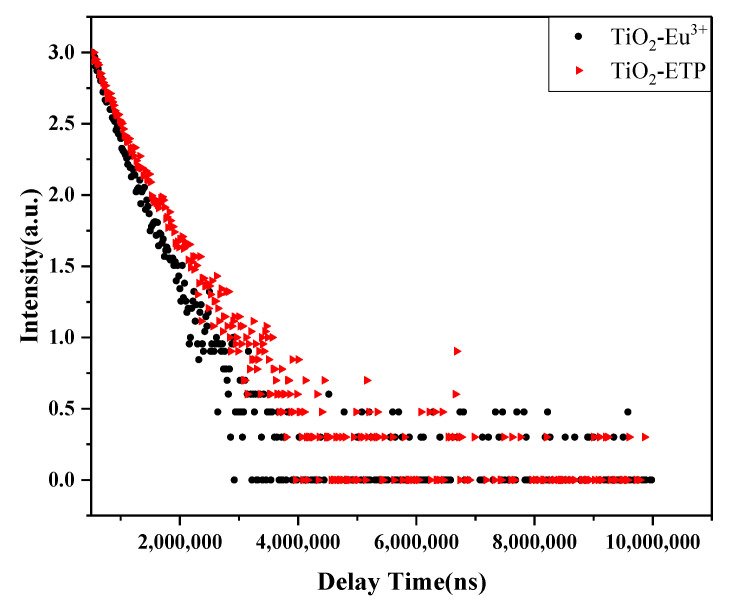
The fluorescence lifetime diagram of TiO_2_-Eu^3+^ and TiO_2_-ETP.

**Figure 10 nanomaterials-11-03283-f010:**
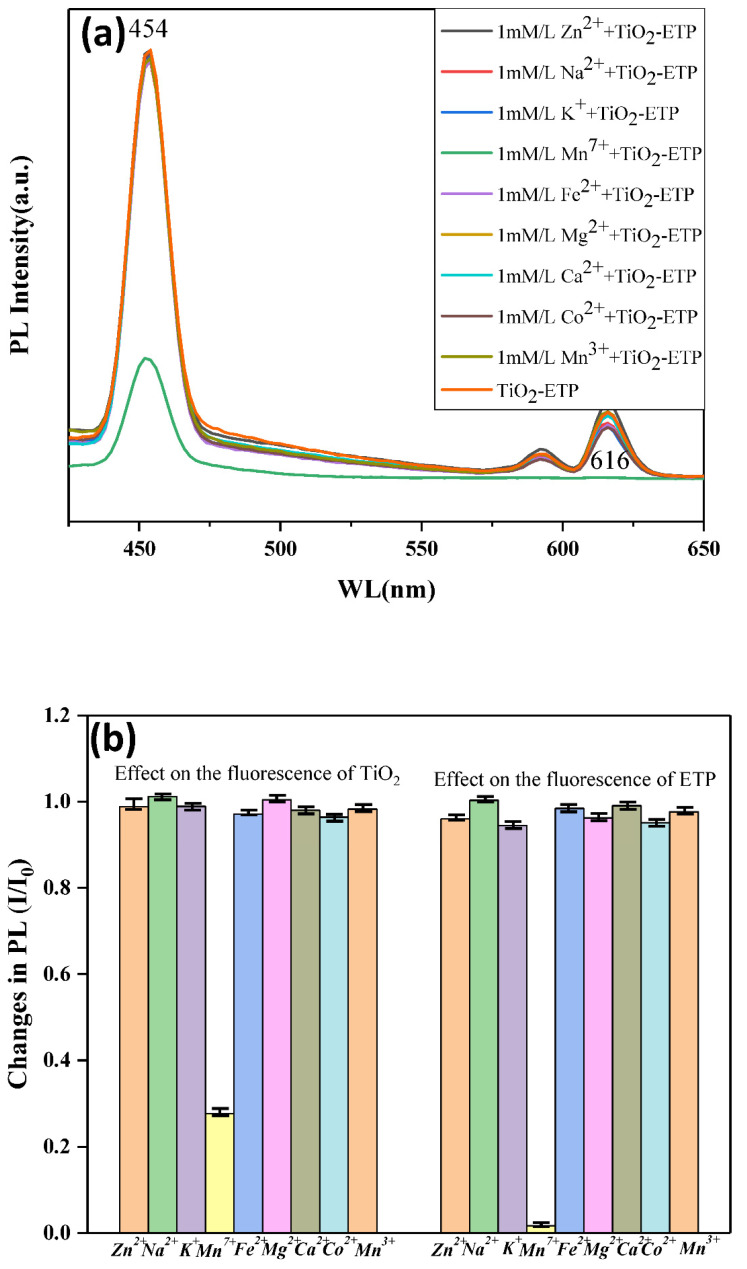
(**a**) Fluorescence spectra of TiO_2_-ETP with metal ions, (**b**) fluorescence intensity ratio (I/I_0_) of the TiO_2_-ETP in an aqueous solution containing no metal ions and a solution containing a single metal ion, (**c**,**e**) fluorescence intensity versus cation concentration for the addition of the Mn^7+^ ions, (**d**,**f**) the linear plot of ΔF/F_0_ against the concentration of Mn^7+^.

**Table 1 nanomaterials-11-03283-t001:** XRD results with parameters.

Sample	hkl	2θ (deg)	D (Å)	FWHM (deg)	Mean Grain Size (nm)	Crystal Structure
TiO_2_	101	25.34	3.51	0.671	15.1	tetragonal
	004	37.84	2.37	0.846	15.0	tetragonal
	200	48.07	1.89	0.710	15.1	tetragonal
TiO_2_-Eu^3+^	101	25.31	3.50	1.115	12.2	tetragonal
	004	37.66	2.38	0.786	12.4	tetragonal
	200	47.88	1.89	0.825	12.1	tetragonal
TiO_2_-ETP	101	25.06	3.55	0.966	9.0	tetragonal
	004	37.48	2.39	0.847	9.1	tetragonal
	200	47.76	1.90	0.971	8.9	tetragonal

## Data Availability

All data, models, and code generated or used during the study appear in the submitted article.

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
