# Peer review of "Smart Mn7+ Sensing via Quenching on Dual Fluorescence of Eu3+ Complex-Modified TiO2 Nanoparticles"

_nanomaterials, 2021, doi:10.3390/nano11123283_

Round 1
Reviewer 1 Report
In this manuscript, the authors report on structural and functional characterization of TiO2 nanoparticles, modified by ETP and compare their properties with those of simple TiO2 and TiO2-Eu3+ samples. In addition to the structural characterization of the different types of samples, crucial experimental results of the manuscript are the optical absorption (OA) in the range UV-visible and the PL emission of samples TiO2-Eu3+ and TiO2-ETP as a function of the Eu3+ concentration, from 2% to 8% mol. On the basis of the comparison of these optical features in the two types of samples, and the observed PL quenching in the presence of Mn7+ in solution, the authors suggest the TiO2-ETP as a good selective and sensitive sensor of such a metal cation.
As regards in particular the optical characterization of TiO2-Eu3+ and TiO2-ETP, I wish to stress the following points.
- a) The authors say that the immission of Eu3+ produces defects and, as a consequence, the decreasing of the energy gap between conduction and valence bands, as a function of the increase of Eu3+ concentration. This decreasing is not monotonous in TiO2-Eu3+, where a saturation limit seems to be reached at a 4% Eu3+ concentration, as confirmed also by the PL data. This effect is not present in samples of TiO2-ETP and this different behavior is ascribed by the authors at a so-called “antenna effect” that is not enough clarified. In my opinion, the authors should explain in more detail such an effect, also because they do not cite any reference.
- b) It is reasonable that the immission of Eu3+ produces defects and such defects are expected to have several different structures. In the level scheme reported in fig. 8(f) is reported just one “defect state” but such a state is related to a single defect or is it representative of a variety of defects? The authors, please, clarify this point.
The authors claim that a TiO2-ETP sample can be used as a selective and effective sensor of Mn7+ and report supporting experimental results in fig.10. I have some comments regarding these results:
- a) The parameters of the best-fit curves of figg. d, f and g cannot be reported with the precision of 4 or 5 significant digits, as they are not justified by the errors in the experimental data. This precision makes no sense absolutely. The authors are requested to correct this mistake.
- b) I do not understand why the authors suggest the curve reporting the ratio y1/y2 (fig. 10 g) as a calibration curve of the Mn7+ sensor, in place of the curve reporting the quenching effect on the PL emission at 464 nm (fig. 10 d) that is more progressive and, in addition, allows to observe a larger Mn7+ concentration range. The authors should comment in more detail on this choice that seems to be quite strange.
Author Response
Response Letter
Nov. 19, 2021
Dear Editor and reviewers,
Thank you very much for your advices for our manuscript. We carefully read your comments and suggestions. Following your opinions, we have corrected and modified the manuscript though out the whole text. The red-marked sites in manuscript are the modifications including the related corrections based on the reviewers’ questions, and our own further modifications to improve the descriptions and understanding quality.
The point by point answers are list as below. Please check and let me know if any further works need to do.
Thank you very much!
With the best regards!
Jianguo Tang
Questions-Answers:
Q1: The authors say that the immission of Eu3+ produces defects and, as a consequence, the decreasing of the energy gap between conduction and valence bands, as a function of the increase of Eu3+ concentration. This decreasing is not monotonous in TiO2-Eu3+, where a saturation limit seems to be reached at a 4% Eu3+ concentration, as confirmed also by the PL data. This effect is not present in samples of TiO2-ETP and this different behavior is ascribed by the authors at a so-called “antenna effect” that is not enough clarified. In my opinion, the authors should explain in more detail such an effect, also because they do not cite any reference.
Answer: Thank you very much for your critical question. You are right, “This decreasing is not monotonous in TiO2-Eu3+, where a saturation limit seems to be reached at a 4% Eu3+ concentration, as confirmed also by the PL data.”, whereas in ETP the decreasing is monotonous. Herein, as experimental data are true. The reason for this difference, can attribute to the “solubility limit” of Eu3+ and ETP in TiO2 host. The former Eu3+ is from EuCl3·6H2O, in which both the counter ion Cl1- and crystalline H2O molecules are typical polar species and affect the solubility of Eu3+. It is solvable at low content, for example, lower than 4%. However, ETP is a complex with organic ligand molecules of 1,10-phenanthroline (Phen) and 2-thenoyltrifluoroacetone (TTA), the organic ligands modified the solubility of ETP. Solubilized ions have efficient interaction with TiO2 host to change the electron transition bandgap[50]. Also, the influences affect the fluorescence behaviors as shown in Fig. 8.
So, we added the explanation into the text in page 10, as below:
The understanding for this deference, can attribute to the “solubility limit” of Eu3+ and ETP in TiO2 host. The former Eu3+ is from EuCl3·6H2O, in which both the counter ion Cl1- and the bringing H2O molecules are typical polar species and affect the solubility of Eu3+. However, ETP is a complex with rich organic ligand molecules of 1,10-phenanthroline (Phen) and 2-thenoyltrifluoroacetone (TTA), the organic ligands modified the solubility of ETP. Solubilized ions have efficient interaction with TiO2 host to change the electron transition bandgap[50]. Also, the influences affect the fluorescence behaviors as shown in Fig. 8.
[50] D. Singh, N. Singh, S.D. Sharma, C. Kant, C.P. Sharma, R.R. Pandey, K.K. Saini, Bandgap modification of TiO2 sol–gel films by Fe and Ni doping, Journal of Sol-Gel Science and Technology 58(1) (2011) 269-276.
Q2: It is reasonable that the immission of Eu3+ produces defects and such defects are expected to have several different structures. In the level scheme reported in fig. 8(f) is reported just one “defect state” but such a state is related to a single defect or is it representative of a variety of defects? The authors, please, clarify this point.
Answer: Thank you very much for your good question. We modified the single line for “defect state” into multi-line mark for the possible multi-defect states.
Q3: The parameters of the best-fit curves of fig g. d, f and g cannot be reported with the precision of 4 or 5 significant digits, as they are not justified by the errors in the experimental data. This precision makes no sense absolutely. The authors are requested to correct this mistake.
Answer: Thank you very much for your great comments. You are right, so we changed as below: We analyzed the error of the experimental data, the error is one digit after the decimal point, so we changed the coefficient of the formula in the figure to one digit after the decimal point. The inset to Fig. 10 (d) shows a graph of the variation of the radiation intensity (I / I0) of TiO2 at 464 nm as a function of the concentration of Mn7+. Linear regression equation: y1 = 20.7C + 23570.9. In the Fig. 10 (f), the linear regression equation for Mn7+ is y2 = 13.8C + 2882.3 (R2 = 0.98, n = 14). The corresponding corrections have been put into page 17.
Q4: I do not understand why the authors suggest the curve reporting the ratio y1/y2 (fig. 10 g) as a calibration curve of the Mn7+ sensor, in place of the curve reporting the quenching effect on the PL emission at 464 nm (fig. 10 d) that is more progressive and, in addition, allows to observe a larger Mn7+ concentration range. The authors should comment in more detail on this choice that seems to be quite strange.
Answer: Originally,we hope to indicate the different effects of Mn7+. However, as you pointed out, we realize that it is easy to make confuse, so we deleted the Fig. 10 g.

Reviewer 2 Report
The paper focuses in the development of Europium complex-modified TiO2 nanoparticle as sensitive Mn (VII) sensors. The nanoparticles have been characterised via TEM, XRD, Raman, XPS, UV-vis and PL spectroscopy, fluorescence lifetime.
The paper falls within the scope of the journal and might be of interest for its readers and the wider scientific community. The manuscript is well-organised and is easy to follow. The conclusions are supported by the data. My comments are shown below:
- Figures 10 a and b are difficult to read
- A closer comparison with related systems described in the literature would be essential
- What about the colloidal stability of the nanoparticles?
- Can the authors suggest a mechanism why the sensors are selective only against Mn (VII)
- What is the performance of the sensors in the presence of interfering molecules?
- Does the temperature affect the sensing performance?
Author Response
Response Letter
Nov. 19, 2021
Dear Editor and reviewers,
Thank you very much for your advices for our manuscript. We carefully read your comments and suggestions. Following your opinions, we have corrected and modified the manuscript though out the whole text. The red-marked sites in manuscript are the modifications including the related corrections based on the reviewers’ questions, and our own further modifications to improve the descriptions and understanding quality.
The point by point answers are list as below. Please check and let me know if any further works need to do.
Thank you very much!
With the best regards!
Jianguo Tang
Referee: 2
Q1: Figures 10 a and b are difficult to read
Answer: Thank you very much. We have made corresponding changes to the font size and legend of Figure 10, which can be better read and understood.
Q2: A closer comparison with related systems described in the literature would be essential
Answer: Based on this situation, the new method to be developed is necessary. Europium complexes constitute an important class of optical probes, with applications ranging from sensing of bioactive species, high throughput assays and screening protocols in vitro, to time-resolved imaging studies in cellulo or in vivo[12]. This provides a possible chance for Mn7+ sensing. So, it becomes possible that we design a strategy to combine both the advantages of Eu3+ fluorescence property and TiO2 dispersion in water medium in this work. Our fluorescent nanomaterials can sense in water or other liquids, with strong anti-interference ability and low cost.
[12] Shashi, Pandya, Junhua, Yu, David, Parker, Engineering emissive europium and terbium complexes for molecular imaging and sensing, Dalton Transactions (2006).
We have mofified on page 2 and 3 as below:
Unlike the previously reported detection using Eu (TTA) 3phen or TiO2, the detection range of nano-material TiO2-ETP is larger, and the detection accuracy and sensitivity are higher.
Q3: What about the colloidal stability of the nanoparticles?
Answer: Thank you very much for your good question. The colloidal nanoparticles generated from TiO2, TiO2-Eu and TiO2-ETP are surely stable for more than 48 hours. So we have enough time to measure. More easily, once the cloudy occurs of their “solutions”, the ultrasonic pulverization is effective to make unique “solution”.
Q4: Can the authors suggest a mechanism why the sensors are selective only against Mn (VII)
Answer: Thank you very much for the nice question. Here in this work, we compared the possible ions, Zn2+, Mn3+, K+, Mn7+, Fe2+, Mg2+, Ca2+, and Co2+ ,as shown in Fig. 10 b, only Mn7+ is effective quencher. For the possible mechanism, actually it needs additional work to confirm. Herein what we can suppose may be: (1) the Mn7+ probably absorbs the similar wavelength light energy with ETP or TiO2, i.e., sharing absorption with ETP; (2) the Mn7+ group may form excimers with TiO2-ETP. We will do further work to confirm in the future works after this publication.
Q5: What is the performance of the sensors in the presence of interfering molecules?
Answer: Thank you very much for your nice critical question. In this work, the possible interferring ions have been tested in Fig. 10. On the other hand, for interfering molecules, we have previously reported the interfering data of organic molecules on Eu3+ complex fluorescence in different host, and the results showed there are no quenching effects occurrence on Eu3+ fluorescence, such as carbohydrates, cholesterol, amino acids and other molecules. [40][58].
[40] B. Su, S. Wang, W. Yang, Y. Wang, L. Huang, K.C. Popat, M.J. Kipper, L.A. Belfiore, J. Tang, Synthesis of Eu-modified luminescent Titania nanotube arrays and effect of voltage on morphological, structural and spectroscopic properties, Materials Science in Semiconductor Processing 113 (2020).
[58] Z. Song, J. Wang, J. Liu, X. Wang, J. Tang, Eu3+-Induced Polysaccharide Nano-Dumbbell Aggregates (PNDA) as Drug Carriers to Smartly Report Drug Concentration through Variable Fluorescence, Sensors and Actuators B Chemical 336(10) (2021) 129724.
Q6: Does the temperature affect the sensing performance?
Answer: Thank you very much for your good question. You are right. Like other fluorescent samples, surely, temperature affects luminescence performance, and therefore, affects sensing. However, due to the detection usually at room temperature, in this work, the sensing detectios are at room temperature.

Reviewer 3 Report
The manuscript is for the preparation of a new luminescent titania nanoparticulate material that is capable of sensing Mn7+ ions at very low concentrations. The paper is well written in general but there are a few corrections required as well as whole details of the testing procedure missing.
Line 39 - the sentence 'Because excessive manganese ions have serious
pollution problems' needs to be rephrased and include a bit more about the pollution problem. Reference 3 does not have details about these.
Line 131 and 135 - Give more details about the procedure for :
- The ultraviolet diffuse reflectance spectrum was obtained using PerkinElmer Lambda 750s (USA).
- Edinburgh Instruments FLS 1000
(America) was used to record the excitation and emission spectra of each sample.
Line 211 - rephrase for the correct meaning 'ETP is successfully is modified on the surface of TiO2 nanoparticles'.
Figure 6 - Figure caption is poorly labelled and is confusing.
Line 338 - No details are given for the method with the concentrations and the procedure for testing the sensing ability of the metal ions.
Were the powdered samples mixed with metal ion solutions and how were these tested?
Author Response
Response Letter
Nov. 19, 2021
Dear Editor and reviewers,
Thank you very much for your advices for our manuscript. We carefully read your comments and suggestions. Following your opinions, we have corrected and modified the manuscript though out the whole text. The red-marked sites in manuscript are the modifications including the related corrections based on the reviewers’ questions, and our own further modifications to improve the descriptions and understanding quality.
The point by point answers are list as below. Please check and let me know if any further works need to do.
Thank you very much!
With the best regards!
Jianguo Tang
Referee: 3
Q1: Line 39 - the sentence 'Because excessive manganese ions have serious
pollution problems' needs to be rephrased and include a bit more about the pollution problem. Reference 3 does not have details about these.
Answer: We have made changes:
In hospital, Mn7+ has widely been used as strong disinfectant, but its strong oxidation property and its heavy metal characteristic make it be a toxic and carcinogen species in water recycle system and human health [3]. In industry, Mn7+ has been used as strong oxidation agents to generate huge amount of toxic waste water[4]. Thus, Mn7+ has attracted a lot of attention among the pollutants in recent years. The excessive manganese ions have made serious pollution to cause toxic drinking water and to damage to plants[5].
[3] J. Crossgrove, Z. Wei, Manganese toxicity upon overexposure, Nmr in Biomedicine 17(8) (2004) 544-553.
[4] Q. Zhong, D.X. Liao, L.X. Ming, Review of Research in the Treatment of Electrolytic Manganese Waste water, China's Manganese Industry (2005).
[5] T. Horiguchi, Mechanism of Manganese Toxicity and Tolerance of Plants, Journal of Plant Nutrition 11(3) (1988) 235-246.
Q2: Line 131 and 135 - Give more details about the procedure for :
- The ultraviolet diffuse reflectance spectrum was obtained using PerkinElmer Lambda 750s (USA).
- Edinburgh Instruments FLS 1000 (America) was used to record the excitation and emission spectra of each sample.
Answer: Thank you very much for your suggestion. We have modified in manuscript page 5, as below:
The ultraviolet absorption spectrum was obtained using PerkinElmer Lambda 750s (USA) with solid sample frame, on which the powder samples were flattened when the powder samples were used. The PL spectrum is an important tool for determining the luminescent properties of materials. Edinburgh Instrument FLS 1000 (UK) was used to record the excitation and emission spectra of each sample, on which the data of excitation spectra, emission spectra, fluorescence lifetimes were collected. The excitation spectra and emission spectra of TiO2 nanoparticles were recorded using the Edinburgh Fluorescence Spectrophotometer FLS 1000 (UK). A 450W xenon arc lamp capable of emitting a continuous spectrum with greater intensity was used as the light source. The excitation monochromator was used to select the specified excitation spectrum with the excitation wavelength of 394 nm. Fluorescence analyzer calibration is a work of high precision requirements, we were in accordance with the instrument operating procedures for rigorous standard sample, sample preparation and processing. Judging from the calibration curve, the expected ideal state is achieved.
Q3. Line 211 - rephrase for the correct meaning 'ETP is successfully is modified on the surface of TiO2 nanoparticles'.
Answer: Thank you very much for the good question. Here, we make a confuse description, and is not proper. So, we deleted this bad sentence, on page 8.
Q4. Figure 6 - Figure caption is poorly labelled and is confusing.
Answer: Thank you very much! We have made change: Fig. 6. (A)overview XPS spectra, (B)XPS spectra of Ti2p, (C) XPS spectra of Eu3d, (D,E,F) XPS spectra of O1s
Q5. Line 338 - No details are given for the method with the concentrations and the procedure for testing the sensing ability of the metal ions. Were the powdered samples mixed with metal ion solutions and how were these tested?
Answer: Thank you very much for this good question. We add the concentration and procedure into experimental section on page 4. TiO2-ETP(0.1mol/L)was added into different testing metal ion solutions with the concentration of 1mM/L, and the mixtures were stirred for 2 hours. Whereas, to sense the quenching behavior of Mn7+, its concentration was set in the range of 1µM/L to 1000µM/L gradiently.
Q6. Were the powdered samples mixed with metal ion solutions and how were these tested?
Answer: Thank you very much! Please check the answer of Q5.

Round 2
Reviewer 1 Report
In the revised version of the manuscript, the authors reply satisfactorily to my questions and remarks, so the present version of the manuscript deserves publication on Nanomaterials.
Author Response
Thank you so much for your comment!
Reviewer 2 Report
The revised manuscript is suitable for publication
Author Response
Thank you so much for your comment!
Reviewer 3 Report
The revisions are fine. Figure 6 DEF caption, mark the figure with the appropriate sample.
Author Response
Thank you so much for your comment.
We have changed figure 6 DEF caption, marked the figure with the appropriate sample:
Fig. 6.(A) overview XPS spectra, (B)XPS spectra of Ti2p, (C) XPS spectra of Eu3d, (D) XPS spectra of O1s in TiO2,(E) XPS spectra of O1s in TiO2-Eu3+, (F) XPS spectra of O1s in TiO2-ETP.